# SARS-CoV-2 and Influenza Co-Infection: Fair Competition or Sinister Combination?

**DOI:** 10.3390/v16050793

**Published:** 2024-05-16

**Authors:** Narasaraju Teluguakula, Vincent T. K. Chow, Mirazkar Dasharatharao Pandareesh, Venkatesha Dasegowda, Vidyasagar Kurrapotula, Shivaramu M. Gopegowda, Marko Radic

**Affiliations:** 1Adichunchanagiri Institute of Medical Sciences, Adichunchanagiri University, Mandya 571448, Karnataka, India; pandareesh@acu.ac.in (M.D.P.); drvenkateshad@bgsaims.edu.in (V.D.); drvidyasagar@bgsaims.edu.in (V.K.); mgshivaramu@bgsaims.edu.in (S.M.G.); 2Department of Microbiology, Immunology and Biochemistry, College of Medicine, University of Tennessee Health Science Center, Memphis, TN 38163, USA; mradic@uthsc.edu; 3Infectious Diseases Translational Research Program, Department of Microbiology and Immunology, Yong Loo Lin School of Medicine, National University of Singapore, National University Health System, Singapore 119228, Singapore; micctk@nus.edu.sg; 4Department of Biochemistry, Adichunchanagiri School of Natural Sciences, Adichunchanagiri University, B.G Nagara 571448, Karnataka, India

**Keywords:** influenza, SARS-CoV-2, COVID-19, co-infection, inflammation, lung pathology

## Abstract

The COVID-19 pandemic remains a serious public health problem globally. During winter influenza seasons, more aggressive SARS-CoV-2 infections and fatalities have been documented, indicating that influenza co-infections may significantly impact the disease outcome of COVID-19. Both influenza and SARS-CoV-2 viruses share many similarities in their transmission and their cellular tropism for replication in the human respiratory tract. However, the complex intricacies and multi-faceted dynamics of how the two pathogens interact to ensure their survival in the same lung microenvironment are still unclear. In addition, clinical studies on influenza co-infections in COVID-19 patients do not provide conclusive evidence of how influenza co-infection mechanistically modifies disease outcomes of COVID-19. This review discusses various viral as well as host factors that potentially influence the survival or synergism of these two respiratory pathogens in the infected lung microenvironment.

## 1. Introduction

Since its emergence in December 2019, the devastating COVID-19 pandemic has culminated in over 7.0 million fatalities globally, according to a World Health Organization report [1]. Although SARS-CoV-2 caused disease throughout the years from 2020 to 2022, significant increases in hospitalizations and mortalities were reported worldwide during winter influenza (flu) seasons. This may be attributed to climatic conditions that are more favorable for virus transmission and/or the possibility of co-infections with other circulating respiratory pathogens. Among the various respiratory pathogens that cause co-infections, influenza viruses have been identified as a significant risk in exacerbating lung disease in COVID-19. SARS-CoV-2 belongs to the coronavirus family and is related to the 2003 severe acute respiratory syndrome (SARS) and Middle East respiratory syndrome (MERS) coronaviruses. Structurally, both SARS-CoV-2 and influenza viruses are enveloped, single-stranded RNA viruses, and their RNA genome is encapsidated by nucleoprotein. While SARS-CoV-2 contains a positive-sense, single large RNA genome, influenza viruses possess a negative-sense segmented RNA genome. Both viral pathogens share similar modes of transmission and infection sites in the respiratory tract [2,3]. Although both viruses exhibit different entry mechanisms, simultaneous infections and the continuous presence of these two pathogens in the same lung environment may result in serious complications and clinical illness. It is believed that SARS-CoV-2 can cause recurring infections, and based on the epidemiology of seasonal and pandemic influenza outbreaks, future viral epidemics are unavoidable. Thus, it is essential to understand the interplay between SARS-CoV-2 and influenza viruses (especially during winter flu seasons) to minimize morbidities and mortalities. This review provides an overview of influenza and SARS-CoV-2 co-infections and discusses how influenza and SARS-CoV-2 viruses compete for survival within the lung microenvironment.

## 2. Impact of Influenza Epidemic Season on SARS-CoV-2 Infection and Disease Outcome

Patients with COVID-19 develop early symptoms such as high fever, cough, myalgia, and dyspnea, but most infected patients recover from mild or subclinical infections. However, fewer than 10% of COVID-19 patients, especially older adult patients, suffer from severe lung disease, which necessitates admission into an intensive care unit (ICU), assisted ventilation, and other supportive therapies. A significant number of these severely ill patients eventually develop progressive clinical manifestations of acute respiratory distress syndrome (ARDS) and succumb due to impaired oxygen intake capacity and multi-organ dysfunction [4]. Pre-existing chronic illnesses, co-morbidities, other health conditions, and microbial co-infections have all been associated with increased fatalities among critically ill COVID-19 patients [5,6,7,8].

The phenomenon of co-infection involves different viral, fungal, or bacterial species in varying combinations [9,10,11,12]. In view of the co-existence of the ongoing and evolving pandemic of SARS-CoV-2 infections with annual flu seasons, several studies have reported that influenza co-infections exacerbate the severity of COVID-19, leading to worse disease outcomes [9,13,14,15,16,17,18,19]. Kim et al. [13] have highlighted that patients with co-infections of influenza and COVID-19 exhibit more severe symptoms and higher mortality rates compared to those infected with COVID-19 alone. In a retrospective study, increased mortalities were observed in influenza co-infected patients compared to adenoviral or *Chlamydia* co-infections among COVID-19 patients [14]. Similarly, co-infection with influenza A virus substantially increased the risk of ICU admission and invasive mechanical ventilation in COVID-19 patients compared to either of the viruses alone [15,16]. Another retrospective study observed an increased risk of long-term illness and poor outcomes in patients co-infected with influenza B viruses during early 2020 [17]. Similarly, reports from China and Iran documented that over 50% of severely ill hospitalized COVID-19 patients were positive for influenza co-infections [18,19]. Influenza co-infections significantly increase in-hospital mortalities compared to SARS-CoV-2 mono-infection or other viral or bacterial co-infections [20]. A clinical study revealed increased mortalities among hospitalized patients with co-infections compared to infection with either of these pathogens alone, thus indicating a lethal synergism between the two respiratory viruses [21]. These findings underscore the importance of monitoring and managing co-infections, particularly during influenza seasons, to mitigate the impact of COVID-19 disease outcomes.

Interestingly, a study on the prevalence of influenza and SARS-CoV-2 co-infections reported to the US Centers for Disease Control (CDC) surveillance system between 2020 and 2022 described more severe disease and mortalities among young patients (<18 years) with influenza co-infections [22]. This raises concerns about the underuse of antivirals or influenza vaccines among younger populations. A recent clinical report identified avian influenza (H5N1) co-infection with SARS-CoV-2—despite severe pneumonia, the patient recovered following treatment with antiviral drugs together with standard care [23]. Multiple clinical reports also documented no differences in clinical signatures and unaltered disease severities among patients with influenza co-infections compared to SARS-CoV-2 infection alone [24,25]. These findings raise questions on how these two viruses interact for survival and indicate dichotomous effects in influenza co-infection. There is a need to understand how these pathogens interact and modulate lung pathophysiology and immune signatures while sharing the same pulmonary microenvironment. Furthermore, due to high levels of similarity in clinical presentations and immunopathologic mechanisms between influenza and COVID-19, the detection of these viral pathogens is essential for the assessment of disease progression and for effective treatment to prevent hospital-associated deaths.

## 3. Alveolar Epithelium: A Common Target for Influenza and SARS-CoV-2 Replication

Both SARS-CoV-2 and influenza viruses can target different epithelial cells in the upper and lower respiratory tract. The infection of the alveolar epithelial lining directly correlates with disease severity in both viral infections. Structurally, the alveolar lining is covered by two types of epithelial cells, i.e., alveolar type I (AT-I) and type II (AT-II) pneumocytes. AT-I cells are squamous and thin membranous cells important for gas exchange and fluid homeostasis, while AT-II cells are small, cuboidal-shaped cells that produce surfactant proteins and lipids. Both AT-I and AT-II cells are permissive of influenza and SARS-CoV-2 infections that induce cytopathic damage [26,27,28], leading to alveolar disintegration, the loss of surfactant, and the impairment of gas exchange. The loss of alveolar integrity subsequently compromises the structural integrity of the extracellular matrix present in the underlying interstitium, and also causes endothelial injury in the capillary network, ultimately culminating in the collapse of the alveolar architecture. The loss of gas exchange due to alveolar damage contributes to overexuberant inflammation and cytokine storm, which eventually progresses into pathologic manifestations of ARDS [4].

Despite their similar cell tropisms, these viruses utilize different routes of entry into the target cells. The influenza virus undergoes endocytosis via the interaction of its hemagglutinin with sialic acid receptors on the epithelial membrane. SARS-CoV-2 gains cell entry via its spike binding to angiotensin-converting enzyme-2 (ACE2) receptors. The spike–receptor binding triggers protease-mediated cleavage of the spike protein by the transmembrane serine protease 2 (TMPRSS2), causing the fusion of the viral membrane with the host cell membrane and subsequent endocytosis of the virus. The initial infection and replication of both influenza and SARS-CoV-2 in AT-II cells trigger the induction of interferon (IFN) and other cytokines by surrounding cells as part of the innate immune responses against the virus at the early phase of infection. These early inflammatory responses damage adjacent AT-I cells, as well as capillary endothelial cells within the walls of the thin alveolar epithelial interstitium, which further mobilizes more inflammatory immune cells to the infection site. The injury and disruption of the alveolar epithelial lining subsequently lead to pathologic alterations of hyaline membrane formation, which is a major characteristic feature identified in both severe influenza and SARS-CoV-2 infections. Interestingly, the replication of SARS-CoV-2 occurs at a slower rate compared to the influenza virus in tracheobronchial and AT-II cells [29]. Influenza virus infection exhibits strong viral antigen staining at 48 h post-infection, whereas SARS-CoV-2 replicates at a slower rate and lower magnitude. The differences in their replication kinetics could significantly impact cellular infectivity and pathogenesis when both viruses share the same microenvironment [29].

## 4. Influenza and SARS-CoV-2 Co-Infection: Competition or Sinister Alliance?

The synergism that occurs during influenza and SARS-CoV-2 co-infection is not yet completely understood. Two possible scenarios of this co-infection can be envisaged, i.e., (i) secondary SARS-CoV-2 infection following primary influenza exposure; and (ii) secondary influenza infection following primary SARS-CoV-2 infection. In both scenarios, the co-existence of both pathogens in the same lung compartment is possible. This may allow both viruses to either compete for their survival in the phenomenon of competitive viral suppression or to exhibit synergism which may cause exacerbated lung pathology (Figure 1).

The competition for survival between these viruses may depend on the time of infection, virus inoculum size, and interactions between virus and host-related entities. Typically, the replication of one virus may inhibit the replication of another virus through competitive suppression. Interestingly, analyses by mathematical modeling of viral co-infections allow us to hypothesize that viruses with faster replication rates will likely suppress viruses with slower replication rates when competing for survival. SARS-CoV-2 has relatively slower replication compared to influenza virus; thus, influenza co-infection could reduce SARS-CoV-2 replication [29]. Several mechanisms have been proposed regarding how influenza can inhibit the replication of SARS-CoV-2. Type I IFN genes (IL-1α and IL-1β) are activated upon recognition of viral nucleotides by pattern recognition receptors (PRRs) such as toll-like receptors (TLRs) or retinoic acid-inducible gene I (RIG-I). This triggers antiviral effects via JAK-STAT-pathway-mediated expression of IFN-stimulated genes (ISGs) followed by synthesis and release of protein kinase R (PKR), Mx proteins, and 2′,5′oligoadenylate synthetase (OAS), which degrade viral RNA [30]. In vitro infection of Vero cells with influenza virus can block SARS-CoV-2 replication kinetics via the induction of type I and type III IFNs [25]. In validation of these findings, hamsters co-infected with SARS-CoV-2 and influenza virus displayed diminished SARS-CoV-2 loads by 5 days post-infection, whereas influenza viral replication remained high, thus indicating the suppression of SARS-CoV-2 in a co-infection scenario. Influenza-induced type I IFN also induces apoptosis in surrounding epithelial cells via Fas ligand signaling [31,32], thus limiting access to ACE2 receptors for invading SARS-CoV-2 [33]. Similarly, prior SARS-CoV-2 infection may restrict influenza virus superinfection via the induction of AT-II cell apoptosis or NOD-like receptor-mediated pyroptosis [34]. Indeed, cells undergoing pyroptotic death aid in eliminating infected cells and thus restrict the replicative niche and survival of the opportunistic intracellular pathogens [34]. In a different instance, competition may occur within the same target cell. If both viruses infect the same epithelial cell, they may compete at different stages of virus replication, including endocytic entry, viral genome replication, assembly, and the release of progeny viruses. One study revealed that influenza infection downregulates the expression of ACE2, thereby preventing SARS-CoV-2 entry and replication [35]. In addition, non-structural genes expressed by influenza and SARS-CoV-2 play a critical role in evading the host’s innate immune response to facilitate their infection of susceptible cells [36,37]. The SARS-CoV-2 NSP1 protein suppresses the expression of IFN-α signaling by attenuating STAT1 and STAT2 phosphorylation; NSP1 also downregulates the IFN-β protein, IFN-λ1, and IL-8 by binding to 18S rRNA of the 40S ribosome in the host cell [37,38].

On the other hand, SARS-CoV-2-influenza co-infection may aggravate the disease’s outcome. Prior influenza infection has been linked to the upregulation of ACE2 receptor expression and enhanced SARS-CoV-2 infectivity in A549 lung epithelial cells [39]. Supporting this finding, an over 5-fold increase in SARS-CoV-2 infectivity was observed in A549, Calu-3 human lung adenocarcinoma epithelial cells and normal human bronchial epithelial cells when pre-infected with influenza virus, thus suggesting the lethal synergism of co-infections in various cell types. In another study, a 2- to 3-fold elevation in ACE2 mRNA levels was observed in influenza-infected A549 cells. However, co-infection with SARS-CoV-2 resulted in an over 28-fold increase in ACE2 mRNA expression, thus suggesting positive feedback from ACE2 expression in a co-infection scenario that contributes to the increased infectivity of SARS-CoV-2 in lung epithelial cells [40]. Interestingly, the augmented SARS-CoV-2 infectivity observed in prior influenza-infected A549 cells was completely blocked when ACE2 expression was knocked down, thus suggesting that the induction of ACE2 is crucial for aggravated SARS-CoV-2 infectivity during co-infection [39].

Another mechanism of lethal synergism was linked to the activation of the type-1 membrane-bound protease furin, which activates influenza hemagglutinin protein and cleaves the SARS-CoV-2 spike protein, thereby facilitating the entry of both viruses into susceptible host cells [41,42]. The activity of furin increases during influenza infection; thus, prior influenza infection may augment predisposition to SARS-CoV-2 during flu season. Likewise, SARS-CoV-2 superinfection at 48 h after a prior influenza challenge in mice exacerbated lung pathology compared to SARS-CoV-2 infection alone [39]. In another study, sequential infection with influenza 4 weeks after SARS-CoV-2 infection in ferrets led to increased clinical signs and inflammation responses compared to influenza infection alone [43]. This increase in susceptibility was even observed among animals challenged with mild SARS-CoV-2, thus indicating the heightened risk of influenza during COVID-19. Similarly, hamsters co-infected with influenza A virus plus SARS-CoV-2 exhibited increased loss of body weight, prolonged pneumonia, and increased disease severity compared with animals infected with either pathogen alone. Interestingly, immunohistochemical analysis of co-infected hamster lungs did not show the co-localization of SARS-CoV-2 and influenza virus, suggesting that in the infected pulmonary environment, these viruses compete for survival to inhibit each other, or that both viruses replicate in adjacent areas in the infected lungs. In addition, the elevation of IL-6 levels in co-infected animals compared to those infected with either pathogen alone suggest the involvement of IL-6 in aggravated lung pneumonia [44]. Immunopathologic analysis of lungs of ferrets co-infected with influenza and SARS-CoV-2 showed extensive inflammation in the nasal cavity and lungs compared to infection with either of these viruses alone [45]. Furthermore, influenza virus, but not SARS-CoV-2, displayed high viral shedding and transmission by direct contact with co-housed animals. In a case report, influenza and SARS-CoV-2 co-infection in a 32-year-old man culminated in severe lung pneumonia. Despite receiving standard care together with the administration of an anti-IL-6 monoclonal antibody and an antiviral agent against influenza, the patient succumbed to the infection due to respiratory failure, indicating a poor prognosis for this co-infection [46]. Overall, the ability of the influenza virus to aggravate SARS-CoV-2-mediated pathogenesis and infectivity suggests that influenza co-infections may constitute a major target in reducing SARS-CoV-2-related morbidities and mortalities in future COVID-19 outbreaks.

Human rhinovirus (HRV), human parainfluenza (HPI), and respiratory syncytial virus (RSV) are common human respiratory viruses that circulate throughout the year. Several clinical reports have identified co-infection with these respiratory pathogens in severely ill COVID-19 patients. To understand the impact of these respiratory pathogens on SARS-CoV-2 infectivity, a study was conducted using lung epithelial cells co-infected with different respiratory viruses together with SARS-CoV-2. Interestingly, the co-infection of alveolar epithelial cells with HRV, HPI, and RSV did not affect SARS-CoV-2 viral loads, although these viruses displayed active replication during co-infection. Lethal synergism was only observed when SARS-CoV-2 was co-infected with the influenza virus, suggesting that influenza co-infection could have a detrimental effect on epithelial survival by exacerbating epithelial injury with subsequent lung damage [39]. In a clinical study, an increase in cough and dyspnea was observed in patients co-infected with SARS-CoV-2 and HRV compared to patients infected with either virus alone. Furthermore, more ICU admissions and deaths were observed among patients with co-infection than those with HRV mono-infection [47]. A recent study revealed that co-infection with RSV enhances the severity of SARS-CoV-2 omicron infection in young children. These findings warrant detailed investigations into how RSV and SARS-CoV-2 co-infections impact the disease’s outcome in children [48].

## 5. Immunopathology: A Common Mediator of Acute Lung Injury in a Co-Infection

Aggravated lung injury from co-infections may exert a cumulative effect on virus-inflicted cytopathic effects together with immunopathology. Severe infection with both influenza and SARS-CoV-2 elicits extensive inflammatory cellular and cytokine responses, which play a central role in the disruption of the thin alveolar epithelial–capillary barrier. Golden Syrian hamsters co-infected with influenza and SARS-CoV-2 displayed extensive inflammatory cell influx, vascular injury, and pulmonary edema compared to hamsters with single virus infection [49]. In another study, increased lung pathology was observed in hamsters co-infected with influenza and SARS-CoV-2 compared to those challenged with either pathogen alone. Elevated IL-6 levels were observed in co-infected animals, suggesting that IL-6-mediated signaling may be involved in greater disease severity [44]. Furthermore, animals infected with SARS-CoV-2 24 h before influenza virus infection displayed severe clinical signs, body weight loss, extensive alveolitis, perivascular and peribronchial inflammatory cellular influx, heightened inflammation, and acute lung injury. In contrast, prior influenza infection ameliorated SARS-CoV-2 infectivity and lung injury.

An excessive lung influx of neutrophils has been observed in both severe influenza and SARS-CoV-2 infections. Upon their recruitment to the infected lung microenvironment, neutrophils exhibit dysregulated activity and generate neutrophil extracellular traps (NETs), which are composed of DNA decorated with toxic components such as histones, neutrophil elastase (NE), and cathepsins [50]. The released NETs function like a double-edged sword, damaging the alveolar capillary bed, triggering epithelial and endothelial necrosis and injuring the pulmonary vasculature. Furthermore, the released extracellular histones activate platelet aggregation to trigger microvascular thrombosis [51,52]. In a co-infection scenario, these insults may precipitously intensify and cause a collapse of the alveolar capillary architecture, leading to the clinical development of ARDS. Furthermore, the damaged alveolar capillary barrier may aid in the systemic dissemination of these viruses and facilitate the adhesion and colonization of opportunistic bacterial or fungal pathogens [53,54]. Influenza and SARS-CoV-2 co-infection may pose a recurring global problem during flu seasons; hence, it is imperative to elucidate the underlying mechanisms of this co-infection to explore and design novel therapeutic interventions.

In addition, influenza and SARS-CoV-2 co-infection in K-18-hACE2 mice resulted in not only increased and prolonged inflammatory cellular infiltrations in infected lungs, but also lymphopenia, reduced levels of IgG, and decreased neutralizing antibody titers in the blood. Interestingly, these co-infected mice also exhibited impaired CD4+ T cell responses against both viruses [55]. These findings highlight that co-infection may impact protective–adaptive immunity, leading to the subsequent long-term impairment of immune responses. Future studies are warranted to investigate whether immunopathogenesis in influenza-SARS-CoV-2 co-infection can exert the cumulative effects of aggravated innate immune response together with impaired adaptive immune response.

## 6. Does Vaccination against Influenza Decrease the Risk of COVID-19?

Annual influenza vaccination drastically reduces ICU admissions among elderly and high-risk individuals, especially during flu seasons in many countries. Influenza immunization also decreases hospitalization and pneumonia caused by other viral or bacterial infections [56,57]. Several clinical studies indicate that prior influenza vaccination diminishes susceptibility to SARS-CoV-2 infection and reduces the burden of COVID-19 by improving clinical outcomes. In a retrospective cohort study, prior influenza vaccination was linked to significant reductions in COVID-19-related hospitalizations, length of stay in the ICU, and the need for invasive respiratory support via mechanical ventilation when compared to unvaccinated individuals [58]. Notably, influenza vaccination of ferrets conferred significant protection to animals infected with influenza alone or those co-infected with influenza and SARS-CoV-2, suggesting that influenza vaccination could help in preventing SARS-CoV-2 disease severity, particularly during flu seasons [44]. Although the precise mechanisms of this protection are still unclear, there is a possibility of bystander-protective effects through the induction of cell-mediated immune responses. One hypothesis is that the restoration of Th1 cell immunity and a decrease in cytokine storm could alleviate SARS-CoV-2 infections [59]. In addition, prior flu vaccination may also trigger non-specific innate immune responses that may aid in the early clearance of SARS-CoV-2. Although specific antibodies and T cell-mediated responses are generated against specific antigens in the classical immune response, the induction of pro-inflammatory responses may provide non-specific immunity and protection against non-specific and unrelated pathogens [59]. Quadrivalent inactivated influenza vaccines can elicit a strong cytokine response following stimulation of immune cells, thus supporting research finding that seasonal influenza vaccination can mitigate SARS-CoV-2 infection [60,61]. Interestingly, the intranasal vaccination of hamsters with an influenza-vectored COVID-19 vaccine that contains NS1-deleted influenza virus carrying the receptor binding domain of SARS-CoV-2 (dNS1-RBD) conferred broad-spectrum protection against challenges with different SARS-CoV-2 variants [62]. The combination vaccine decreased viral loads and attenuated pro-inflammatory cytokines (IL-6, IL-1β, and IFN-γ), thus ameliorating immunopathology and tissue injury after SARS-CoV-2 infection. These findings are important, as COVID-19 disease severity is due to a combination of virus-inflicted tissue damage together with immunopathology mediated by overactive immune cells and excessive inflammatory cytokines. Notably, in one study, protection by prior influenza vaccination could protect against influenza and SARS-CoV-2 co-infection. However, prior immunity against SARS-CoV-2 does not protect in this co-infection scenario. The protection by influenza vaccination has been associated with neutralizing antibodies, but not CD4+ and CD8+ T cells [63]. Future studies are warranted to decipher the mechanisms underlying how prior influenza immune responses help to alleviate influenza/SARS-CoV-2 co-morbidity. Bacille Calmette-Guérin (BCG) vaccine is prepared from a live attenuated strain of *Mycobacterium bovis.* BCG vaccination can elicit heterologous immune responses that are effective against cancers (such as bladder cancer) by reducing tumor progression and cancer recurrence. Heterogeneous protective effects against SARS-CoV-2 infections were also conferred by influenza and BCG vaccinations [64]. More experimental evidence is needed to better understand the mechanisms underlying how heterogeneous immunity can prevent or mitigate SARS-CoV-2 pathophysiology.

## 7. Conclusions and Future Directions

The risk of influenza co-infections during the COVID-19 pandemic remains unclear. Notwithstanding this, the dramatic increase in SARS-CoV-2 infections and fatalities during influenza epidemic seasons supports the notion that co-infection with both viruses exacerbates lung injury. Both influenza and SARS-CoV-2 infections give rise to complex cytopathic and immunologic reactions in the lungs and other organs. These reactions may aggravate lung and organ pathogenesis, or these viruses may compete and inhibit each other for their survival in the pulmonary micro-compartments. In view of the overlapping and similar clinical manifestations of both viral infections, virus-specific tests remain pivotal for diagnostic confirmation, e.g., reverse-transcription polymerase chain reaction and multiplex assays. Given that both viruses are likely to continue to co-circulate every year, it is critical to continue effective surveillance and to characterize novel variants of SARS-CoV-2 and influenza viruses [65,66]. During forthcoming winters, a double viral threat is anticipated that may require the administration of COVID-19 vaccines together with seasonal influenza vaccines to alleviate the respiratory disease burden, especially among vulnerable populations. Additionally, the protective effects of influenza vaccination against severe outcomes of SARS-CoV-2 infection suggest a promising avenue for reducing the burden of COVID-19. Future vaccination strategies should explore potential cross-protection conferred by influenza vaccines, while also elucidating the heterogeneous effects of different vaccination approaches. Non-pharmacological interventions such as mask wearing, social distancing, and hand hygiene remain pivotal measures for mitigating the transmission of both viruses [67]. As the world grapples with evolving respiratory virus infections, future research endeavors should unravel the complexities of co-infection, guide surveillance strategies, and inform the targeted interventions crucial for optimizing public health responses during influenza epidemic seasons and their impact on SARS-CoV-2 infection and disease outcomes.

## Figures and Tables

**Figure 1 viruses-16-00793-f001:**
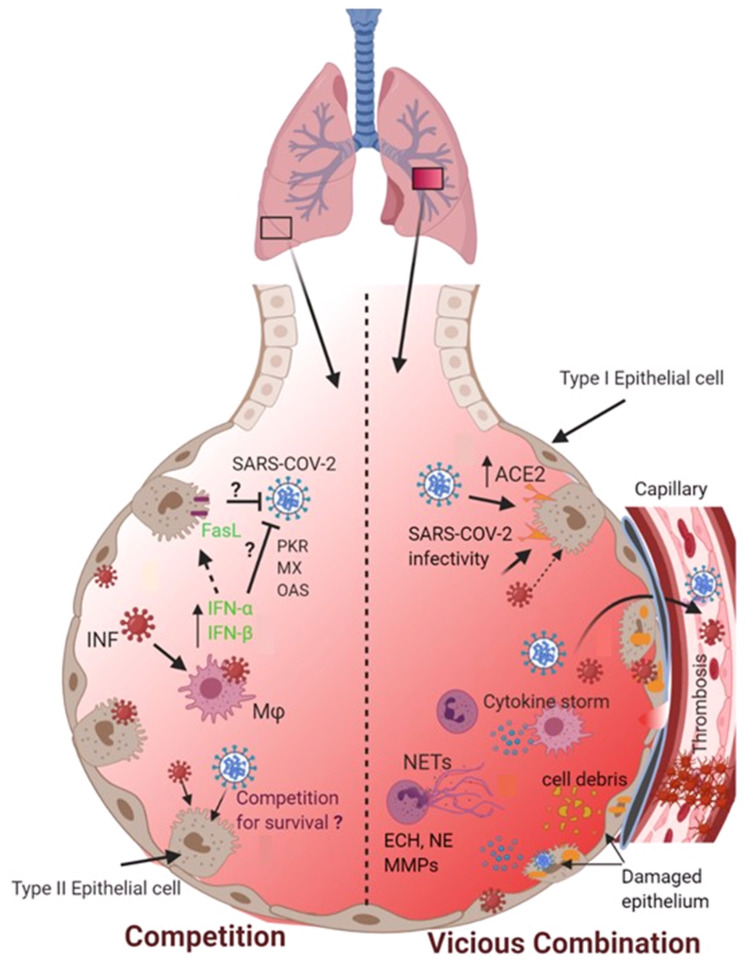
The potential fate of co-infection of SARS-CoV-2 and influenza viruses in the pulmonary alveolar microenvironment. Two possible scenarios may occur when both SARS-CoV-2 and influenza viruses encounter and co-infect the lungs. On the one hand, co-infection may cause these two viral pathogens to compete for their survival. On the other hand, co-infection could result in a vicious combination that may exacerbate lung pathophysiology by causing alveolar capillary damage, overwhelming inflammatory responses, the release of NETs, and the formation of pulmonary vascular thrombosis.

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
