# Peer review of "SARS-CoV-2 and Influenza Co-Infection: Fair Competition or Sinister Combination?"

_viruses, 2024, doi:10.3390/v16050793_

Round 1

Reviewer 1 Report

Comments and Suggestions for Authors

The presented review is of undoubted value and interest for scientists and students working in the field of virology and can be published after minor corrections and clarifications.

Carefully checking all references recommended. For example, in link #2 the year is incorrect (2013 instead of 2023).

In addition, authors need to check the correctness of the references. The article (ref. 2) is devoted to modeling COVID-19 seasons, but not to study of infection sites in the upper and lower respiratory tract.

It is also advisable to adhere more strictly to the sequence of references, avoiding unnecessary returns to the same works (for example, ref. No. 24).

Author Response

Reviewer 1

Thank you for your valuable comments.

Carefully checking all references recommended. For example, in link #2 the year is incorrect (2013 instead of 2023).

Response: As suggested, all the references have been carefully checked and corrected.

In addition, authors need to check the correctness of the references. The article (ref. 2) is devoted to modeling COVID-19 seasons, but not to the study of infection sites in the upper and lower respiratory tract.

Response: We edited and included another reference in the revised manuscript.

It is also advisable to adhere more strictly to the sequence of references, avoiding unnecessary returns to the same works (for example, ref. No. 24).

Response: We have corrected the sequence of the references.

Reviewer 2 Report

Comments and Suggestions for Authors

The authors address an exciting problem of virus co-infections and their consequences focusing on SARS-CoV-2 and influenza, albeit other respiratory viruses such as respiratory syncytial virus, human rhino- and parainfluenza viruses are also mentioned. Several aspects are discussed: The disease outcomes; Virus competition vs. sinister alliance; Immunopathology; Importance of vaccination. 

The coverage of the factual material is wide thus the Review is rather interesting and sure warrant publication. It is clear at the moment that the molecular and cellular mechanisms of these co-infections are yet unclear. It is also obvious that there is not enough data yet to draw conclusions about specific problems. However, the important issues are specified, and future directions are outlined. 

One point was not clear enough for me.  In section "Impact of influenza epidemic season on SARS-CoV-2 infection and disease outcome" the authors claim about "a marked increase in SARS-CoV-2 infections and fatalities during the winter flu seasons of 2020 and 2021" but provide only one reference #9 which actually does not confirm this claim. (It is the third paragraph at page 2). Instead, as far as I know those flu seasons (especially the 2020 winter season inthe Northern Hemisphere when the lockdown was announced in many countries) showed not the increased but decreased morbidity compared to the previous (ordinary) flu seasons. I would like to see more facts, more references and more reasonable conclusions.

There are several missprints that should be corrected/ specified:

Page 3, 16th line from the bottom: "in both severe influena AND (should be inserted) SARS-CoV-2"

Page 6, 6th line from the bottom: " "IL-8 mediated". Did the authors really mean IL-8 or it is a missprint (IL-6 is actually meant)?

Page 6, line 21st line from the bottom: Please decipher the abbreviation ICU here (at the moment it is decophered laler at page 7.

Page 6, line 17th line from the bottom: Do the authors mean ref. 43 or ref. 17?

Page 8: Please decipher the abbreviation BCG in line 9 (at the moment it is deciphered below in line 13).

Author Response

Reviewer 2

Thank you for your valuable comments.

One point was not clear enough for me.  In section "Impact of influenza epidemic season on SARS-CoV-2 infection and disease outcome" the authors claim about "a marked increase in SARS-CoV-2 infections and fatalities during the winter flu seasons of 2020 and 2021" but provide only one reference #9 which actually does not confirm this claim. (It is the third paragraph at page 2). Instead, as far as I know those flu seasons (especially the 2020 winter season in the Northern Hemisphere when the lockdown was announced in many countries) showed not the increased but decreased morbidity compared to the previous (ordinary) flu seasons. I would like to see more facts, more references and more reasonable conclusions.

Response: This section has been rewritten for more clarity as follows: “In view of the co-existence of the ongoing and evolving pandemic of SARS-CoV-2 infections together with annual flu seasons, influenza co-infections may have a significant exacerbating impact on the disease outcome of COVID-19.9 Further, the co-existing COVID-19 pandemic along with seasonal influenza also highlights the need for effective surveillance to monitor co-circulation of SARS-CoV-2 and influenza viruses in future.”

There are several missprints that should be corrected/ specified:

Response: The typographical errors are thoroughly corrected in the revised manuscript.

Page 3, 16th line from the bottom: "in both severe influena AND (should be inserted) SARS-CoV-2"

Response: The correction is included in the revision

Page 6, 6th line from the bottom: " "IL-8 mediated". Did the authors really mean IL-8 or it is a missprint (IL-6 is actually meant)?

Response: The correction to IL-6 is included in the revised manuscript.

Page 6, line 21st line from the bottom: Please decipher the abbreviation ICU here (at the moment it is decophered laler at page 7.

Response: The abbreviation is corrected.

Page 6, line 17th line from the bottom: Do the authors mean ref. 43 or ref. 17?

Response: The reference has been corrected.

Page 8: Please decipher the abbreviation BCG in line 9 (at the moment it is deciphered below in line 13).

Response: The abbreviation is corrected.

Reference #9 has been changed to: Singer BD. COVID-19 and the next influenza season. Sci. Adv. 2020; 6, eabd0086.

Response: The reference is corrected in the revised manuscript